# Licon: A Diverse, Controllable and Challenging Linguistic Concept Learning Benchmark

**Shenglong Yu[1]  Ying Zhang[1] *  Wenya Guo[1]  Zhengkun Zhang[1]  Ru Zhou[1]  Xiaojie Yuan[1]**
[1] College of Computer Science, VCIP, TMCC, TBI Center, Nankai University, China

{yushenglong,guowenya}@dbis.nankai.edu.cn
{yingzhang,yuanxj}@nankai.edu.cn
{zhangzk2017,zhouru}@mail.nankai.edu.cn

## Abstract

*Concept Learning* requires learning the definition of a general category from given training examples. Most of the existing methods focus on learning concepts from images. However, the visual information cannot present abstract concepts exactly, which struggles the introduction of novel concepts related to known concepts (e.g., 'Planet'→'Asteroids'). In this paper, inspired by the fact that humans learn most concepts through linguistic description, we introduce *Linguistic Concept Learning* benchmark (**Licon**), where concepts in diverse forms (e.g., plain attributes, images and text) are defined by linguistic descriptions. The difficulty to learn novel concepts can be controlled by the number of attributes or the hierarchical relationships between concepts. The diverse and controllable concepts are used to support challenging evaluation tasks, including concept classification, attribute prediction, and concept relationship recognition. In addition, we design an entailment-based concept learning method (EnC) to model the relationship among concepts. Extensive experiments demonstrate the effectiveness of EnC. The benchmark will be released to the public [1].

## 1 Introduction

*Concept Learning* aims to learn the definition of a general category from given information (Han et al., 2019; Ye et al., 2020a). This task formulates the innate human ability of generalization in terms of concepts, i.e., the ability to quickly learn novel concepts from few information based on known concepts. Prior work mainly focuses on *Visual Concept Learning* (Mao et al., 2019; Han et al., 2019; Li et al., 2020; Mei et al., 2022), discovering concepts from the visual information (e.g., images). However, the images can only display concrete information via limited examples, such as colors and

---

*Corresponding author.
[1] https://github.com/yushenglong1/Licon

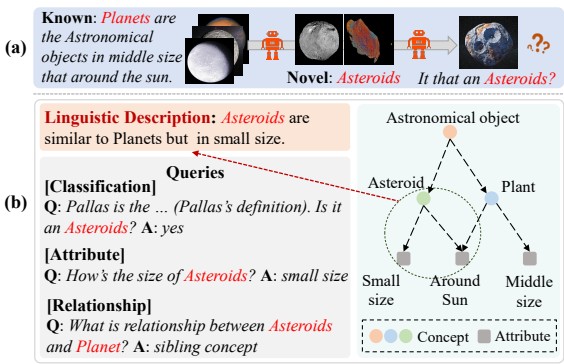

Figure 1: An example of concept learning. Based on the known concept 'Planets', it is difficult to recognize the novel concept, 'Asteroids', according to several images. However, a simple linguistic description helps to associate 'Asteroids' with known 'Planets'. Then concept learner can answer questions about Asteroids.

shapes, while abstract concepts cannot exactly presented in the visual contents. As shown in Figure 1 (a), due to the similar appearance, it is difficult to recognize the relationship between 'Planets' and 'Asteroids' according to the images separately.

By contrast, humans can learn various concepts through language (Lake et al., 2015), especially in academic areas such as physics and psychology. Compared to images, language can precisely describe the complex relationship between abstract concepts. As shown in Figure 1 (b), the attributes of 'Asteroids' can be clearly described as 'Asteroids are similar to Planets but in small size'. Inspired by this fact, we propose to associate novel concepts with known ones through linguistic descriptions, called *Linguistic Concept Learning*. As shown in Figure 2 (b), novel concepts are defined by linguistic descriptions rather than images. The linguistic descriptions formulate various relationships between novel concepts and known concepts for single-modal (Figure 2 (c)) or multi-modal (Figure 2 (b)), leading to a more challenging task.

In this paper, we introduce a linguistic concept

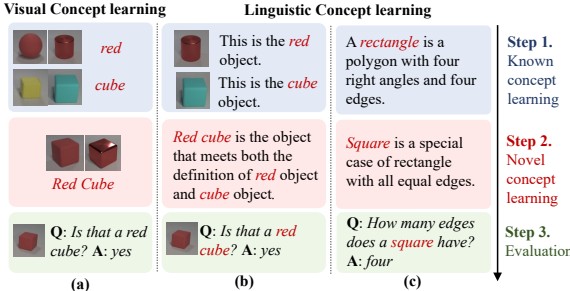

Figure 2: Illustration of visual concept learning and linguistic concept learning. Concept learning process typically consists of three steps: Known concept learning, Novel concept learning, and Evaluation.

learning benchmark (**Licon**). Compared to previous concept learning datasets (Lin et al., 2014; Johnson et al., 2017; Hudson and Manning, 2019), Licon has the following properties: 1) **Diverse**: Known concepts are presented in diverse forms, including plain attributes, concrete visual images and abstract textual definitions, named Licon-A, Licon-I, and Licon-T, respectively. The construction methods are also diverse, which are synthetic about concrete concepts (Licon-A and Licon-I), or artificially designed about real-world abstract concepts (Licon-T). 2) **Controllable**: The synthetic datasets, Licon-A and Licon-I, allow us to control the difficulty of synthetic concepts by changing the number of attributes or relationships among them, which can support testing the effectiveness of concept learners in various scenarios. 3) **Challenging**: Licon requires concept learners to recognize comprehensive information about novel concepts, including categories, attribute values and relationships with other concepts. The diverse concept forms, especially the real-word datasets (Licon-T), and complete evaluation tasks pose great challenges for linguistic concept learning.

In addition, we propose an entailment-based linguistic concept learning method (**EnC**), which uses box embeddings and GNN to explicitly model the relationships between concepts. Further, EnC leverages *concept-level entailment* as loss to optimize the model. EnC achieves the best performance on both synthetic and real-world datasets, severing as a strong baseline for facilitating future work.

In summary, our contributions are:

• We are the first to propose the challenging task, *Linguistic Concept Learning*, and collect a diverse, controllable and challenging benchmark, Licon.

• We develop EnC, an entailment-based model

to capture the relationships (i.e., entailment and contradiction) between concepts.

• We conduct extensive experiments to verify the effectiveness of EnC. Additionally, we also explore the challenges of linguistic concept learning by designing various experiments.

## 2 Related work

**Concept learning.** Concept learning focus on the ability of learning concept through a diverse set of inputs: by looking at images (Mao et al., 2019; Han et al., 2019;Li et al., 2020; Mei et al., 2022), reading sentences describing the properties of objects (Dan et al., 2021;Wang et al., 2019;Ye et al., 2020a); or about the relationships between concepts (Han et al., 2019). However, most of the works focus on visual concept learning, i.e., the process of learning novel concepts relies on visual information. We differ from these works by transforming the learning process to rely only on linguistic descriptions, which allows us to represent novel concepts with a wider distribution in a more flexible way.

**Zero-shot learning.** Recently, due to the development of Large pre-trained language models (LMs) (Devlin et al., 2018;Liu et al., 2019;Raffel et al., 2020), many works (Zhang et al., 2019;Yin et al., 2019;Ye et al., 2020b;Sanh et al., 2021;Mishra et al., 2021) focus on zero-shot learning, which aims to train a model that can classify objects of unseen classes via transferring knowledge obtained from other seen classes with the help of semantic information (Pourpanah et al., 2022). While the experiment setup in our task meets the definitions of zero-shot learning, our work is significantly different from these works as 1) Our benchmark comprises a large set of concepts for learning from linguistic description as opposed to working on a limited set of tasks in prior work. 2) Our benchmark focuses on concept learning, and provides a comprehensive evaluation, including categories, attribute values and relationship about concepts, rather than just classification.

## 3 Dataset Construction

In this section, we describe the collection details of our proposed benchmark, *Licon*. We divided Licon into three datasets, namely Licon-A (synthetic, plain attribute), Licon-I (synthetic, image), and Licon-T (real-world, text) according to the different construction methods and data sources.

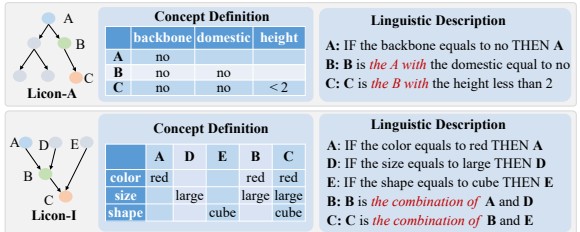

Figure 3: The hierarchical structures, definitions and descriptions of concepts in the synthetic datasets of the proposed *Licon* benchmark, i.e., Licon-A and Licon-I.

## 3.1 Licon-A

To begin with, we collect Licon-A, a single-modal synthetic dataset with controllable difficulty. In Licon-A, the concepts are the fabricated species of the bird, which are represented by plain attributes and have strict hierarchical relationships. We mainly focus on the process of learning novel concepts based on known concepts, and the dataset has universality in the field of concept learning and is not specific to the narrow domain of birds. The example of Licon-A is shown in Figure 10.

We generate Licon-A by first selecting attributes from the pre-defined schema used in (Menon et al., 2022). As shown in Figure 3, concepts are hierarchically generated based on the selected attributes. Specifically, we refer to the concepts that do not depend on other concepts as *base concepts*, such as concept A in Licon-A and concepts A, D, E in Licon-I of Figure 3. Additional concepts are generated by attaching new constraint to the definitions of existing concepts. We use the number of attributes (including 4 and 8) to control the difficulty of learning a concept.

Afterwards, the descriptions are generated based on the definitions of the concepts. As mentioned in Menon et al. (2022) , we use the following types of descriptions to control the linguistic complexity ($v_i$ denotes $i^{th}$ statement that describes the value of an attribute, $c$ denotes the name of concept): 1) Simple: *IF $v_1$ THEN $c$*; 2) Conjunctive: *IF $v_1$ AND $v_2$ THEN $c$*; 3) Disjunctive: *IF $v_1$ OR $v_2$ THEN $c$*. Next, we generate two types of questions for each concept, one focusing on using the learned concepts to identify an object, and the other focusing on the concept itself, including the attribute and the relationship between the concepts.

Finally, we split the concepts into a training set and a test set. All the base concepts are included into the training set as known concepts, while the remaining concepts as well as their descriptions are split proportionally into the training set and the test set. In sum, we generate 6 sub-datasets with a different learning difficulties and description complexity as Licon-A. For more details, please refer to Appendix A.1.

## 3.2 Licon-I

To further explore the linguistic concept learning under multi-modal scenarios, we collect Licon-I. In this dataset, the concepts are the kinds of object present in the image, and the example is shown in Figure 11. As used in Johnson et al. (2017), we choose color, size, shape and material as the basic attributes with their own values. As shown in Figure 3, in contrast to Licon-A, we use combination rather than attachment to generate concepts in Licon-I. Then we also generate two types of questions for each concept and split the concepts in the same way as Licon-A.

In Licon-I, we generate 2 sub-datasets with different concept hierarchies, including 2-layers and 3-layers, which means a concept can be combined by up to two or three base concepts, as shown in Figure 3. More details in Appendix A.2.

## 3.3 Licon-T

We construct Licon-T by collecting linguistic concept learning questions from the National Civil Service Examination of China, which give candidates a linguistic description of the abstract concept and ask them to answer questions about that concept.

We collect raw data at official websites[2] and remove questions containing images or charts. We then remove questions that require external knowledge to answer. Finally, we remove duplicated questions and the resulting dataset contains 1008 pairs. Since the original dataset is written in Chinese, we hired professional English speakers to translate the dataset manually. And we will also release the Chinese version of that dataset. For more details, please refer to Appendix A.3.

## 4 Dataset analysis

In this section, we describe the statistics and data distributions of Licon.

**Dataset Statistics:** Table 1 shows the statistics of Licon. Compared to Licon-A and Licon-I, Licon-T has the most concepts and the fewest questions for each concept to learn, which indicates its difficulty.

---

[2] https://www.chinagwy.org/

|  | Licon-A | Licon-I | Licon-T |
| --- | --- | --- | --- |
| Train concepts | 90 | 280 | 681 |
| Test concepts | 18 | 56 | 227 |
| Avg,# Q/Concept | 36.05 | 39.68 | 4 |
| Classification | 28.05 | 30.22 | - |
| Attribute | 2 | 4.34 | - |
| Relationship | 6 | 7.34 | - |
| Vocabulary | 207 | 713 | 9448 |
| Avg,# tokens/Q | 11.79 | 9.75 | 49.83 |
| Avg,# tokens/Des | 22.24 | 13.15 | 13.1 |

Table 1: Statistics of datasets in our proposed *Licon* benchmark. 'Q' and 'Des' denotes question and description, respectively. 'Classification', 'Attribute' and 'Relationship' are the types of the corresponding questions.

Besides, in real-world datasets, the reasoning steps are complex and difficult to categorize, combining the use of concepts for classification, attributes and relationships of concepts. So the synthetic tasks, Licon-A and Licon-I are created to evaluate the linguistic concept learning task in controlled settings with straightforward descriptions and targeted questions for evaluation. This approach allows us to evaluate the learning ability of the model in detail under different settings.

**Characteristics of Licon-A:** Figure 4 (a) shows the distribution of Licon-A. Licon-A is divided into six sub-datasets based on the number of attributes and the type of description. The description types include simple, conjunctive and disjunctive descriptions, which are used to control the linguistic complexity of descriptions. And the number of attributes is used to control the difficulty of learning concepts through objects. In addition, to diversify descriptions and make Licon-A more similar to real-world datasets, we proportionally append negation (20%) and numerical judgments (40%) as linguistic variations of the descriptions. The examples are shown in Figure 4 (a).

**Characteristics of Licon-I:** Figure 4 (b) shows the distribution of Licon-I. Licon-I is divided into two sub-datasets based on different concept hierarchies. In Licon-I-1, we generate novel concept by combining the features of two base concepts, e.g., *red object + cube = red cube*. And in Licon-I-2, we can combine three base concepts, e.g., *red cube in middle size*. Objects in Licon-I-2 require more attributes (average 2.53) to be judged when classified compared to Licon-I-1 (average 1.81).

**Reading complexity:** We analyze the reading complexity of Licon by applying *Flesch reading ease* [3],

[3] https://en.wikipedia.org/wiki/Flesch

where a low score implies complex sentences. The result is shown in Figure 5 (a). The Licon-A is the easiest to read in terms of complexity, with scores ranging from 60 (easy to understand for 13-year-old's) to 100 (very easy to read for 11-year-old's). Licon-I is similar to Licon-A, but a bit harder. Licon-T is the most complex, its score is concentrated in about 20 (very difficult to read), even including negative values (extremely difficult to read).

**Topic distribution:** We analyze the topic distribution (manually labeled) of Licon-T. As shown in Figure 5 (b), the concepts in our dataset are widely distributed across multiple domains, which demonstrates the practicality of our real-world dataset. The top three fields are psychology, sociology, and economics and the concepts are the terminology used in these fields.

**Analysis of difficulty:** The difficulty of the datasets, namely Licon-A, Licon-I, and Licon-T, varies from easy to hard. Licon-A is the easiest with a clear attribute set. In contrast, Licon-T is the hardest as it contains the most concepts and the least amount of data to learn each concept. The description of Licon-T is particularly challenging to understand, and it also requires strong reasoning abilities to grasp the given options effectively.

## 5 Experiment Setup and Models

### 5.1 Formal Definition

The dataset $D$ for linguistic concept learning is structured as $\{(c_1, q_1, a_1), (c_1, q_2, a_2) \dots (c_1, q_{m_1}, a_{m_1}) \dots (c_n, q_{m_n}, a_{m_n})\}$, where $(c_k, q_s, a_s)$ is an instance form $D$ with $c_k$ denoting the linguistic description of the concept, $q_s$ and $a_s$ denoting the concept related question and its answer. It is worth noting that each concept description $c_k$ is related to multiple questions.

Our goal is to train a model capable of learning novel concepts from linguistic descriptions that explain novel concepts in light of known concepts. We train our model over a set of seen concepts $\{c_1, c_2, ..., c_k\}$ and evaluate generalization to the novel concepts $\{c_{k+1}, ..., c_n\}$. At evaluation, the model is evaluated by the accuracy of answering multiple choice questions related to the novel concepts. Since the model never sees the novel concepts during training, our experiments are performed with zero-shot settings.

Besides, for synthetic datasets at evaluation, we divide the results into two categories *T1* and *T2*,

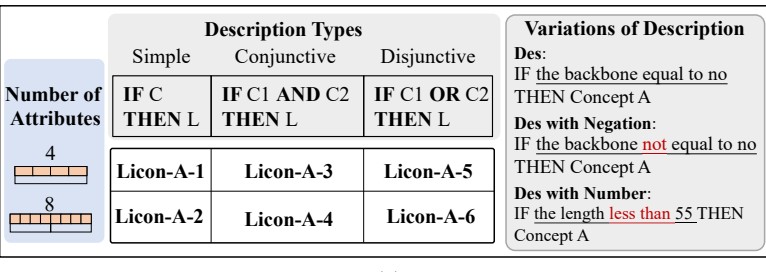

Figure 4: Distribution of sub-datasets in Licon-A and Licon-I. In (b), the 'questions type' is the percentage of questions about classification/attribute/relationship. And 'Avg.#Attribute' denotes the number of attributes used to classify an object.

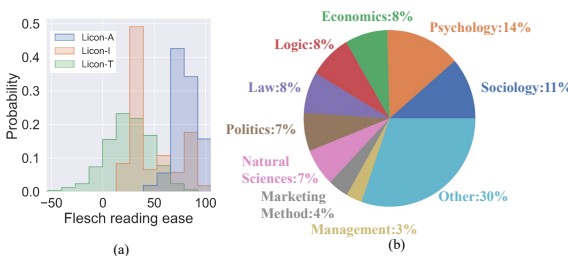

Figure 5: Statistics Distribution of *Licon*. (a) the reading complexity distribution of *Licon*; (b) the topic distribution of Licon-T.

which are designed for evaluating the ability of models to identify category (the former) and other information, including attributes and relationships, of concepts (the latter).

## 5.2 Baseline models

For our baselines, a pre-trained RoBERTa (Liu et al., 2019) model is utilized to better understand the linguistic description. The input of RoBERTa is text, so we need to encode the different modalities as text. For Licon-A, we follow the Menon et al. (2022), encoding the plain attributes as text sequence. We linearize it as a sequence of attribute-value pairs, separated by [SEP] tokens. For Licon-I, we use the pre-trained models, such as ResNet-101 (He et al., 2016) pre-trained on ImageNet (Deng et al., 2009) or CLIP (Radford et al., 2021), to encode the images as embeddings. Then, the embeddings are appended as the prefix to the questions in the same way as Li and Liang (2021). For Licon-T, the data is all textual, we don't need additional manipulation. After transforming all features into text, we combine these features with questions and candidates as the inputs of RoBERTa. Then, RoBERTa is able to answer multiple-choice questions, ignoring different modalities. And We refer to the textual

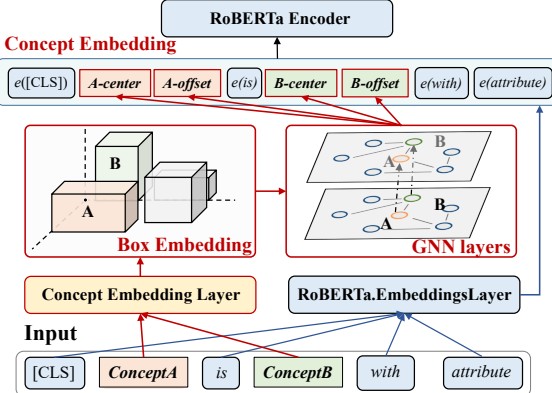

Figure 6: Architecture of our proposed method. We transform the embeddings of concepts in descriptions to box embeddings and further optimize via GNN layers.

format of inputs by 'Questions-as-Text' or 'QaT'.

We use RoBERTa in two ways, in the former we concatenate the concept description and QaT as the input of RoBERTa, noted as *RoBERTa.C*. And in the latter, we separately take the concept description and QaT as input, and compute the final probability by dotting the outputs, noted as *RoBERTa.D*. More details can be found in Appendix B.

## 5.3 EnC

To better associate the knowledge of known concepts with novel concepts, we develop EnC, which uses box embeddings and graph neural networks (GNN) (Hamilton et al., 2017) to model the relationship between concepts and use *the concept-level entailment* loss to optimize the model.

Figure 6 shows the overview of our proposed entailment-based concept learner, EnC. The fundamental idea behind EnC is generating box embeddings as concept embeddings to maintain hierarchical relationships between concepts. The GNN in EnC is used to realize knowledge interaction

|  | 1-T | 2-T | 3-T | 4-T | 5-T | 6-T | Avg.#T | Avg.#T1 | Avg.#T2 |
|---|---|---|---|---|---|---|---|---|---|
| RoBERTa.C | 61.84 | 61.84 | 65.99 | 62.82 | 65.45 | 67.11 | 64.18 | 63.19 | 67.94 |
| RoBERTa.D | 60.53 | 60.53 | 72.11 | 55.13 | 68.18 | 61.84 | 63.05 | 61.83 | 67.99 |
| RoBERTa-MNLI | 67.11 | 67.11 | 78.23 | 64.10 | 70.50 | 67.76 | 69.14 | 68.98 | 70.02 |
| EnC | **78.74** | **77.63** | **85.71** | **75.64** | **77.70** | **77.63** | **78.84** | **80.86** | **72.91** |

Table 2: Experimental results on Licon-A. 'T' is the accuracy on all questions, and 'T1', 'T2' is the accuracy on corresponding questions. '$k$-T' means the accuracy of all questions on A-$k$ dataset. And 'Avg.#X' is the average accuracy of 'X' type questions on all sub-datasets.

|  | 1-T | 1-T1 | 1-T2 | 2-T | 2-T1 | 2-T2 | Avg.#T | Avg.#T1 | Avg.#T2 |
|---|---|---|---|---|---|---|---|---|---|
| CNN-LSTM | 45.83 | 48.37 | 32.77 | 46.52 | 50.47 | 34.31 | 46.18 | 49.42 | 33.54 |
| CLIP-RoBERTa.D | 53.08 | 47.22 | 83.19 | 58.99 | 49.72 | 87.68 | 56.04 | 48.47 | 85.44 |
| ResNet-RoBERTa.D | 53.49 | 48.04 | 81.51 | 60.21 | 51.04 | 88.56 | 56.85 | 49.54 | 74.39 |
| CLIP-RoBERTa.C | 50.89 | 45.42 | 78.99 | 56.99 | 47.06 | 87.68 | 53.94 | 46.24 | 83.33 |
| ResNet-RoBERTa.C | 51.03 | 45.26 | 80.67 | 54.48 | 44.51 | 85.34 | 52.76 | 44.89 | 83.01 |
| ResNet-Enc | **63.61** | **59.31** | **85.71** | **64.31** | **56.17** | **89.44** | **63.96** | **57.74** | **87.58** |

Table 3: Experimental results on Licon-I; 'T' is the accuracy on all questions, and 'T1', 'T2' is the accuracy on corresponding questions. '1-X', '2-X' means the data on I-1 and I-2.

|  | EN | ZH |
|---|---|---|
| RoBERTa.D | 28.71 | 29.21 |
| RoBERTa.C | 32.67 | 30.19 |
| RoBERTa(TE).C | 33.17 | 30.19 |
| EnC | **34.15** | **34.15** |

Table 4: Experimental results on Licon-T

between concepts. The concept-level entailment is used to train box embedding to represent hierarchical relationships.

Firstly, we extend box embedding for knowledge bases (Vilnis et al., 2018) to linguistic concept learning task. Each concept is represented as a box in a high-dimensional space, i.e., the embedding for each concept is a tuple of two vectors $e_c = (Cen_c, Off_c)$. $Cen_c$ is the center of the box and $Off_c$ is the offset in box. Both are in $R^d$, where $d$ is the embedding dimension. By this method, each concept is mapped to a box in the high-dimensional space, whose center is $Cen_c$ with the edge length of $2 * Off_c$.

Then, to model the hierarchy between concepts, we apply GNN on the box embeddings of concepts. We use a designed semantic parser to parse the linguistic description and construct a relational graph $G$ for the known concepts. And for novel concept $c$, we append it as a node in $G$, and initialize it with embeddings $e_{c0} = (Cen_c, Off_c)$, which is the average embedding of known nodes. Then, we input it to the GNN to get $e_{c1} = (Cen_{c1}, Off_{c1})$, where $Cen_{c1} = GNN(Cen_{c0}, G)$ and $Off_{c1} =$

$GNN(Off_{c0}, G)$. In other words, the center of known concept affects the center of novel concept and so does the offset.

We also propose concept-level entailment as loss to better optimize the embeddings of concepts. For a novel concept, we take it as *hypothesis*, and for its hypernym (parent concept in $G$, i.e, expand from that known concept) and cohypernym (sibling concept in $G$, i.e, expand from a same known concept), we treat them as different *premises*. For a premise, we decide whether the premise contains the hypothesis (*entailment*) or contradicts it (*contradiction*).

**For hypernym** The novel concept $c$ with box embeddings $e_c$, should entail its hypernym $c_h$ with box embeddings $e_h$. And then we utilize the feature of box embedding to calculate the overlap values of two boxes $e_c$ and $e_h$, which we note as $l_e$.

**For cohypernym** The novel concept $c$ with box embeddings $e_c$, should contradict its cohypernym $c_o$ with box embeddings $e_o$, in order to distinguish between the two. We also compute the overlap values of two boxes $e_c$ and $e_o$, which we note as $l_o$.

In experiments, we build our model based on RoBERTa. We combine the concept description and QaT as the input and transform the embeddings of concepts in description to box embeddings using the method described above. Finally, we combine $l_e$ and $l_o$ with cross-entropy loss $l$ to form the final loss $L$:

$$L = l - (l_e - l_o).$$

And we train EnC using the final loss. More details will be described in Appendix B.

## 6 Results

### 6.1 Results on Licon-A

We evaluate the performance of models on Licon-A and the results are summarized in Table 2. For Licon-A, we also use RoBERTa pre-trained on MNLI corpus (Williams et al., 2017) as a baseline. In general, EnC outperforms all baselines with an accuracy improvement of about 10% under diverse experimental settings, demonstrating the effectiveness of our concept learning approach. Besides, the EnC improves the performance both on *T1* and *T2* questions. In addition, by comparing the results of sub-datasets under the same description type, i.e, A-1 to A-2, A-3 to A-4 and A-5 to A-6, we find that fewer attributes can lead to higher accuracy.

### 6.2 Results on Licon-I

Our main results on Licon-I are summarized in Table 3. We can draw the following conclusions according to the results. 1) EnC with ResNet as the encoder outperforms the baselines on all types of problems, with an average improvement of 10%. 2) Compared to Licon-A, model on the Licon-I is significantly less accurate, especially for *T1* questions, which indicates the difficulty of multi-modal concept learning and application. 3) Different distribution of question types (shown in Figure 4) affects the focus of model learning, for example, in the pre-trained models, *T2* questions perform better in Licon-I-2 than in Licon-I-1, while the opposite is true for *T1*. 4) The complexity of concepts hierarchy (compared Licon-I-2 to Licon-I-1), which mainly affects the difficulty of object classification (*T1* questions), may be caused by 'Avg.#Attribute' shown in Figure 4.

### 6.3 Results on Licon-T

Finally, we evaluate the proposed method on Licon-T, which is more complex than synthetic datasets. We also test the performance of the model pre-trained on textual entailment corpus. For English, we use MNLI (Williams et al., 2017). And for Chinese, we use CCL2018 [4]. Our main results are shown in Table 4. Overall, EnC achieves the highest accuracy compared to the baselines. However, the performance is still unsatisfactory. The primary reason is the difficulty of understanding complex textual descriptions and the requirement for high-

---

[4]http://www.cips-cl.org/static/CCL2018/call-evaluation.html#task3

level reasoning abilities.We discuss more details in Appendix C.

### 6.4 Ablation experiments

We perform ablation experiments on EnC, including the following variants: **EnC w/o Box** only includes GNN to affect the normal embedding of the concept. **EnC w/o GNN** transforms the normal embeddings in RoBERTa to box embeddings without GNN. **EnC w/o Loss** has the same structure as EnC, but it excludes the external loss to optimize. The results are shown in Table 5, compared to **Enc**, there is a significant drop in accuracy for other variants, which indicates the effectiveness of our model. In addition, we find that using box embedding brings the greatest improvement for EnC.

## 7 Discussion

### 7.1 The Challenge of Negation

Negative sentences are a serious challenge in language understanding. With negation, a sentence can have exact opposite meaning. To identify the challenge of negation in linguistic description, we perform experiments to explore the influence of negation. For each subset in Licon-A, we generate two versions, the former consisting of non-negative sentences and the latter consisting of all negative sentences. We evaluate EnC on two datasets and show the relative gain of the non-negative version compared to the negative version in Figure 7.

Our results indicate that the negation reduces the performance in all conditions. Besides, there exists a similar distribution among the same attribute numbers, which indicates the influence of negation is related to the difficulty of learning concepts through items. And when an object has fewer attributes (A-1, A-3, A-5), the drop is mainly composed in *T1* questions, and mainly in *T2* questions when an object has more attributes (A-2, A-4, A-6).

### 7.2 Visualization of box embedding

We visualize the box embeddings of concepts in Figure 8. Since the different dimensions in a box embedding are independent, we cannot use t-SNE to reduce the dimension, which will lead to changes in the structure of the box embeddings. Therefore, we randomly choose two dimensions of the box embeddings, then draw the edges.

As shown in Figure 8, our proposed loss helps to construct the hierarchical relationship between concepts. In (a), the embedding of the *concept A*

| | 1-T | 2-T | 3-T | 4-T | 5-T | 6-T | Avg.#T | Avg.#T1 | Avg.#T2 |
|---|---|---|---|---|---|---|---|---|---|
| EnC | **78.74** | **77.63** | **85.71** | **75.64** | **77.70** | **77.63** | **78.84** | **80.86** | **72.91** |
| w/o Box | 71.05 | 71.05 | 83.67 | 70.51 | 71.94 | 74.34 | 73.76 | 74.24 | 72.45 |
| w/o GNN | 76.32 | 73.68 | 82.99 | 69.23 | 76.26 | 74.34 | 75.47 | 76.69 | 71.47 |
| w/o Loss | 77.63 | 76.32 | 84.35 | 73.08 | 76.36 | 76.32 | 77.34 | 78.72 | 72.39 |

Table 5: Results of ablation study on Licon-A.

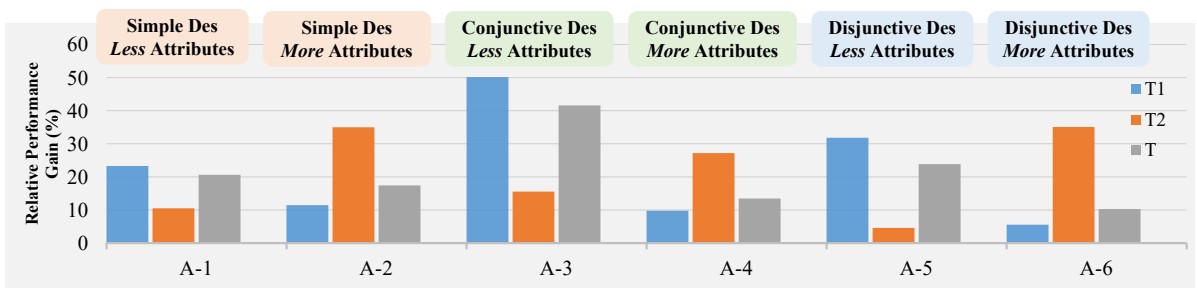

Figure 7: Analysis of the effect of negation. The bars indicate the relative performance gain of the non-negative version over the negative version.

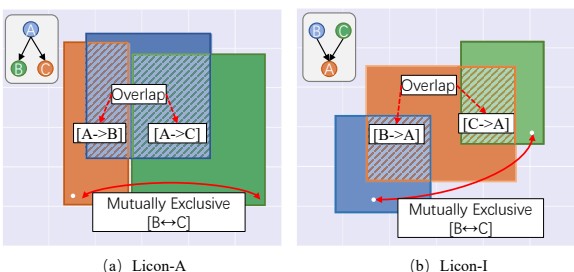

(a) Licon-A     (b) Licon-I

Figure 8: Visualization of box embeddings in Licon-A and Licon-I. The rectangle is the embedding of the concept in corresponding color.

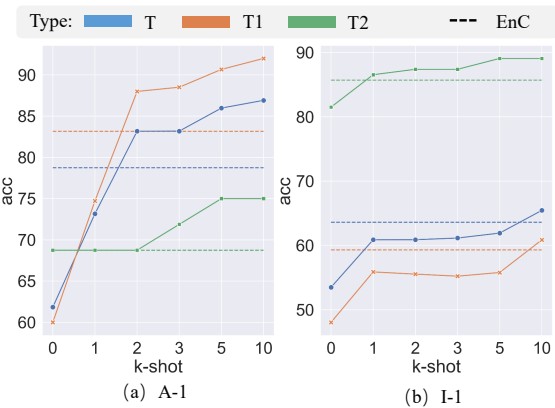

(a) A-1     (b) I-1

Figure 9: Few-shot Learning Performance on Licon-A-1 and Licon-I-1 datasets.

simultaneously covers the embeddings of children. While the embeddings of the cohypernym concepts (B and C) are mutually exclusive. In (b), the *concept A* is combined by two other concepts and this relationship is also shown in the box embeddings.

### 7.3 Effective of few-shot

Our experiments are performed with zero-shot settings. To further explore the few-shot settings of linguistic concept learning, we evaluate the RoBerta.D and ResNet plus RoBerta.D model on A-1 and I-1 datasets, respectively. For each novel concept, we provide external $k$ questions in the training set as *k-shot* conditions.

The results are shown in Figure 9. And the horizontal line indicates the performance of EnC on that dataset. We draw the following conclusions. 1) Few-shot learning, even one-shot learning, significantly improves the performance of the model,

especially when the learning process is difficult (the results on I-1 vs A-1). 2) In general, one-shot learning brings the greatest improvement in all types of problems for both datasets. 3) Compared to baselines, EnC achieves one-shot performance on A-1 and better than five-shot performance on I-1, which indicates the effective of EnC.

## 8 Conclusion

In this paper, we introduce Licon, a diverse, controllable and challenging benchmark for linguistic concept learning, involving plain attributes, images and text. Besides, we develop the EnC, an entailment-based model to capture the hierarchical relationships between concepts. Extensive experiments demonstrate the effectiveness of EnC.

## Limitations

The results on Licon-T are unsatisfactory. One reason is the complexity of linguistic descriptions in the real world. As shown in Figure 5 (a), some descriptions are extremely difficult to read, which indicates that we need to further improve the understanding ability of the pre-trained model in the future.

We plan to introduce curriculum learning to enhance the understanding ability of complex concepts in the future. As mentioned in (Li et al., 2020), the process of human learning involves starting small and gradually building knowledge. Similar to human learning, we'll train the model to learn from easy concepts to complex ones. The planned curriculum includes three stages: (1) simple linguistic concepts, (2) complex options for simple concepts, and (3) more complex concepts. With this training method, we expect better performance on Licon-T. However, limited by the cost of manually annotating the difficulty degree of the concepts, it would require extra significant resources to implement this approach. So we plan to explore this method in the future.

## Ethics Statement

Since the original concept concepts are collected from the National Civil Service Examination of China, there is no private information (e.g., phone, email, postcode and location) in data collection of the real-world dataset, Licon-T. Besides, EnC is trained to recognize the characteristics of the defined concepts, and the trained model will not produce any sensitive information.

## Acknowledgments

The research is supported by the National Natural Science Foundation of China (No. 62272250), the Natural Science Foundation of Tianjin, China (No. 22JCJQJC00150, 22JCQNJC01580), the Fundamental Research Funds for the Central Universities (No. 63231149).

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

# A Additional details of creating Licon

In this section, we will discuss the details of creating Licon.

## A.1 Licon-A

Licon-A is a dataset consisting of text data. And we generate Licon-A by the following steps: 1) Select the table schema and group the attributes into different groups. 2) Generate definitions and linguistic descriptions of the concepts. 3) Generate questions for the concepts and divide the concepts into training and test sets. And the example is shown in Figure 10.

### A.1.1 Table schema

For the table schema, which we used to select attributes, we follow the settings in Menon et al. (2022). For all the attributes in a schema, they define a fixed domain from which values for that attribute can be sampled. And we use the *Species of bird* as our table schema, which is shown in Figure 13, and the target is to classify a bird into a particular species based on attributes. We consider the species of birds as concepts, which are judged by the values of attributes. In table schema, the attributes can be either of types categorical or numeral.

It is worth noting that our task focuses on exploring the ability of the model to learn novel concepts through linguistic descriptions, so unlike Menon et al. (2022), we use only the first table and divide the attributes into groups. Then, we use attributes within groups, which are non-overlapping, to define concepts.

### A.1.2 Creating the definition of concept

To generate the linguistic descriptions and questions for concepts, we first generate the definition of concept, as shown in Figure 3. For a concept, we classify an object by making a judgment on each attribute.

Then, we generate two types of concepts for Licon-A, one for concepts that do not depend on other concepts, called *base concepts* and the other for concepts that are generated by attaching external judgments on attributes to existing concepts.

In addition, the definition also varies on the following axes:
- Hierarchical relationship: $\mathbb{H} = \{2,3\}$
- Negation: $\mathbb{N} = \{$'no negation','with negation'$\}$

We apply $\{0.8, 0.2\}$ as the ratio in the two axes.

### A.1.3 Creating linguistic descriptions

We use a template-based approach to convert the definition into a linguistic description.

We convert the operator in definition as:
- == $\rightarrow$ 'equal to'
- \> $\rightarrow$ 'greater than'
- \>= $\rightarrow$ 'greater than or equal to'
- \< $\rightarrow$ 'lesser than'
- <= $\rightarrow$ 'lesser than or equal to'
- != $\rightarrow$ "not equal to'
- !> $\rightarrow$ 'not greater than'
- !< $\rightarrow$ 'not lesser than'

And for the two types of concepts, the linguistic description is different as shown in Figure 3. One is *IF v THEN c* and the other is *$c_1$ is the $c_2$ with v.* ($v$ denotes conditional statement that describes the value of an attribute, $c$ denotes the name of concept)

### A.1.4 Generating questions for concept

We generate two types of questions for each concept, one about the classification and the other related to the concept itself, including attribute and relationship.

For the classification questions, we randomly generate objects based on attributes and assign labels using definitions. And in order to avoid too simple questions, at most one of the attributes of an object does not meet the definition.

For questions about attributes, we generate questions for each attribute. The options for questions are selected from the fixed domain in the table schema. And if there is *AND* or *OR* in the definition, we will fix one attribute and require the other.

For questions about the relationship between concepts, we randomly choose other concepts, including hypernym and cohypernym concepts, and determine their relationships based on their definitions.

Then, we split these concepts into a training set and a test set and form a sub-dataset.

### A.1.5 Different sub-datasets

Finally, we create 6 sub-datasets by varying along the following axes:
- Structure of description: $\mathbb{S} = \{$'simple', 'conjunction', 'disjunction'$\}$
- Number of attributes: $\mathbb{N} = \{4,8\}$

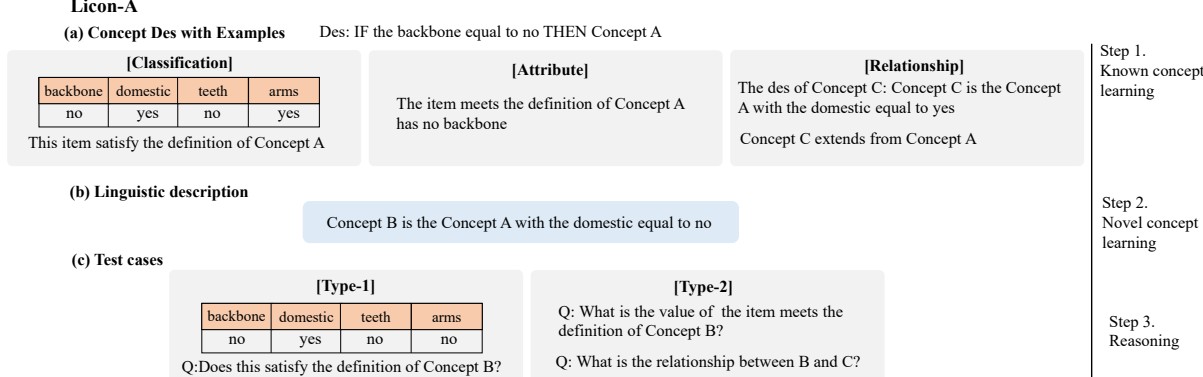

Figure 10: Examples from Licon-A. We first train the model to understand concept **A**, then the model learns concept **B** through a linguistic description and at evaluation, answer questions about the concept **B**.

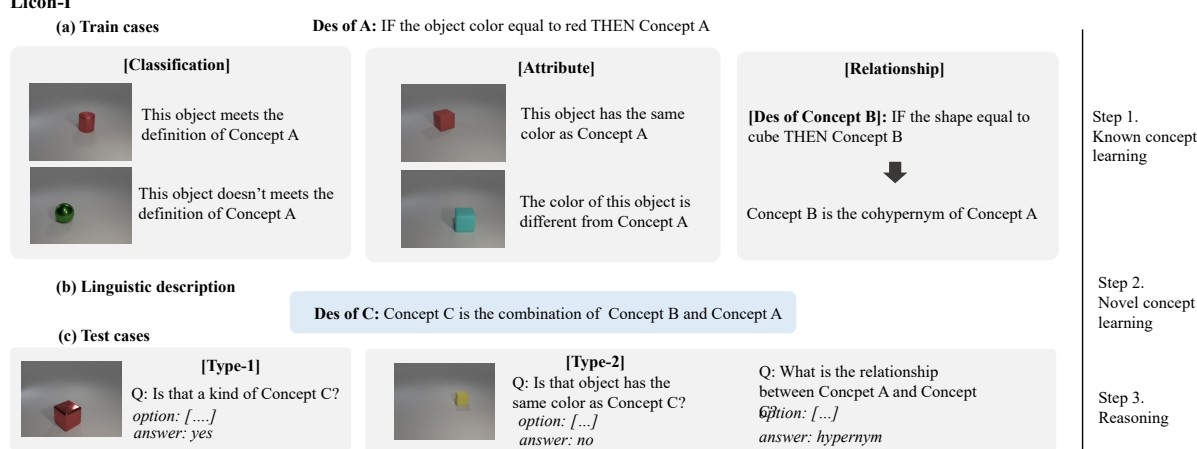

Figure 11: Examples from Licon-I. We first train the model to understand concept **A** and concept **B**, then the model learns concept **C**.

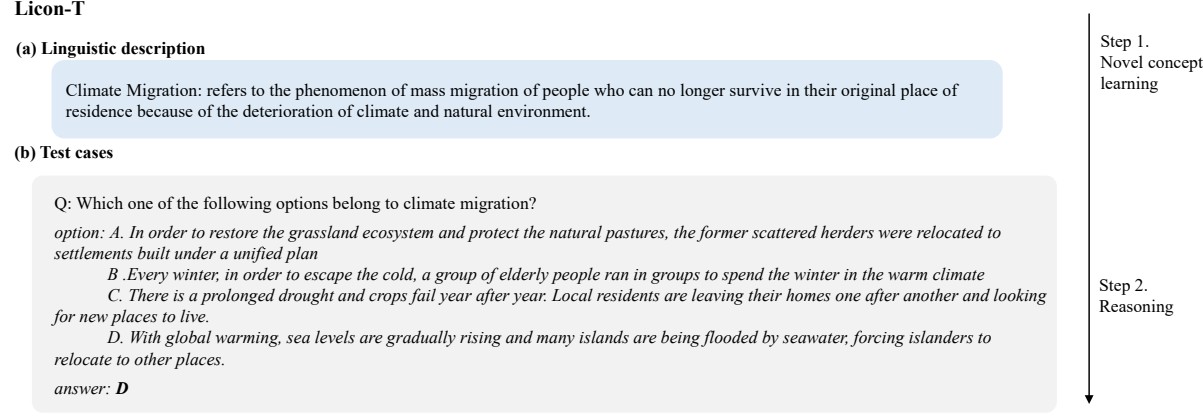

Figure 12: Examples from Licon-T, since the known concepts are stored in the pre-trained model as commonsense, we only train the model to learn novel concepts through linguistic descriptions.

"description": "This dataset is used to predict the type of birds based on the given attributes. Each row provides the relevant attributes of a bird.",
"column_names":{
        "size" : ["categorical", ["large", "medium", "small"]], "size (number)" : ["number", [10, 100]],
        "color" : ["categorical", ["red", "blue", "green", "brown", "pink", " orange", "black", "white"]],
        "head" : ["categorical", ["yes", "no"]], "length" : ["categorical", ["tall", "medium", "short"]],
        "length (number)" : ["number", [10,100]],
        "tail" : ["categorical", ["yes", "no"]],
        "number of faces" : ["number", [1,3]],
        "arms" : ["categorical", ["yes", "no"]],
        "legs" : ["categorical", [2, 4, 6, 8]],
        "hair" : ["categorical", ["yes", "no"]],
        "wings" : ["categorical", ["yes", "no"]],
        "feathers" : ["categorical", ["yes", "no"]],
        "airborne" : ["categorical", ["yes", "no"]],
        "toothed" : ["categorical", ["yes", "no"]],
        "backbone" : ["categorical", ["yes", "no"]],
        "venomous" : ["categorical", ["yes", "no"]],
        "domestic" : ["categorical", ["yes", "no"]],
        "region": ["categorical", ["asia", "europe", "americas", "africas", " antartica", "oceania"]]
        }
}

Figure 13: Table schema of Licon-A: Species of Birds.

"description": " This dataset is used to describe a synthetic object ",
"column_names":{
        "size" : ["categorical", ['large', 'middle', 'small']],
        "color" : ["categorical", ['blue', 'brown', 'cyan', 'gray', 'green', 'purple', 'red', 'yellow']],
        "shape" : ["categorical", ['cube', 'cylinder', 'sphere']],
        "material : [" categorical ",[ 'metal', 'rubber']],
        }
}

Figure 14: Table schema of Licon-I: Species of synthetic objects.

## A.2  Licon-I

Licon-I is a dataset consisting of multi-modal data. We follow the same steps as Licon-A to generate Licon-I. And the example is shown in Figure 11.

### A.2.1  Table schema

For the table schema, we use four attributes to form the table schema, which are color, size, shape and material. And the full table schema is shown in Figure 14.

### A.2.2  Creating the definition of concept

We also generate the definition of concepts in Licon-I as shown in Figure 3. And different from Licon-A, the operators in Licon-I only contains *equal*.

We generate two types of concepts, one for concepts that do not depend on other concepts, *base concepts*, and the other for concepts that are generated by combining two existing concepts.

### A.2.3  Creating linguistic descriptions

The template-based approach is also used in Licon-I. The linguistic description for the two types is different. One is *IF v THEN c* and the other is *$c_3$ is the combination of $c_1$ and $c_2$*, which means that $c_3$ must satisfy both the definition of $c_1$ and $c_2$.

### A.2.4  Generating questions for concept

We also generate two types of questions for each concept in Licon-I. For the classification questions, we generate images and answers following Johnson et al. (2017). And to avoid too simple questions, at most one of the attributes of an object doesn't meet the definition. For questions about attributes, we generate images with the specific attributes as shown in Figure 11. And for questions about the relationship, we perform the same operation as in Licon-A.

### A.2.5  Different sub-data

Finally, we create 2 sub-datasets by changing the hierarchical relationship: $\mathbb{H} = \{2,3\}$, which means the number of levels of all concepts. Under the *2* setting, a concept can be combined with up to two base concepts. And under the *3* setting, a concept can be combined with three base concepts.

### A.3  Licon-T

To create Licon-T, we collected 3142 questions from the Chinese National Civil Service Examination over the past two years. We then hired six annotators to filter out questions involving charts or figures, and questions that can't be answered

solely from the concept descriptions. And the final Licon-T contains 1008 questions. We will release the English and Chinese versions of Licon-T for concept learning study.

As shown in Fig 12, the concepts in Licon-T are defined as "concept name: the corresponding complex description". There is no explicit set of attributes in Licon-T, so it is difficult to classify the descriptions into different types, such as Licon-A and Licon-I, so we take Licon-T as a whole in our experiments.

## B  Training details

In this section, we describe the details of the training, including the parameters of the model and the process of the training model.

### B.1  Details of concept split

For Licon-A, we proportionally split the concepts into training sets, validation sets, test sets, accounting for 60%, 10%, and 30%, respectively. And for Licon-I, because it has more concepts, the ratios are 60%, 20%, and 20%. It is worth noting that during training, *base concepts* function as known knowledge, so such concepts must be in the training set. For Licon-T, we also split the concepts according to 60%, 20%, and 20%.

### B.2  Model parameters

The *RoBERTa.C*, *RoBERTa.D*, *RoBERTa-MNLI* in our paper is the same as the pre-trained RoBERTa-based model available on HuggingFace library. And the *CLIP* is available through Radford et al. (2021). The *ResNet* is used in Johnson et al. (2017). We implement **EnC** based on RoBERTa, with a two-layer GNN to model the hierarchy of concepts.

### B.3  Hyper-parameter settings

All hyper-parameters of the transformer-based models we use are kept default to the settings in the HuggingFace library. For RoBERTa, we use 'roberta-base' as our checkpoint. And for RoBERTa-MNLI, we use 'textattack/roberta-base-MNLI' as the checkpoint. For Chinese dataset in Licon-T, we use 'hfl/chinese-roberta-wwm-ext' as the checkpoint for RoBERTa and CCL2018 corpus for RoBERTa(TE). For EnC, we finetune the checkpoint of RoBERTa pre-trained on textual entailment corpus. For CLIP, we use the 'ViT-B/32' as checkpoint. And for ResNet, we follow the settings in Johnson et al. (2017).

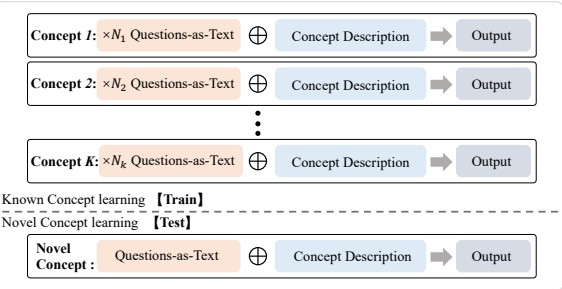

Figure 15: Experimental setup: the model is trained over known concepts and evaluated generalization to novel concepts.

For Licon-A and Licon-I, the max length for RoBERTa-based models is 64. We increase the limit to 256 tokens in Licon-T. We use the AdamW (Paszke et al., 2019) as the optimizer for all models. And we choose $1e - 5$ as the learning rate. The batch sizes are kept as 16.

### B.4  Experimental setup

We train our model on seen concepts $c_1, c_2, ..., c_k$ and evaluate on novel concepts $c_{k+1}, ..., c_n$ as shown in Figure 6. The model is performed with a zero-shot setting. And during training, the model never sees novel concepts and generalizes to novel concepts at evaluation.

### B.5  Training process

**Training on Licon-A and Licon-I:**

Due to the template-based description generation method, descriptions can be easily transformed into the relational graph $G$ of known concepts by a designed semantic parser. Besides, for the known concepts in the training set, we randomly initialize the box embeddings. And optimize the box embeddings of known concepts and parameters of GNN during training. At evaluation, we append novel concepts as a node on the graph based on their relationships and initialize the box embeddings with the average box embedding of all concepts in the training set. Then we get the final box embedding of novel concepts through the GNN based on $G$ and transform the normal word embeddings to box embeddings as the input of RoBERTa.

Furthermore, during training, for an epoch, we first train the model to learn the base concept and then force the model to learn how to understand linguistic descriptions through the remaining concepts in training sets. Finally, we test the model on novel concepts, which give only linguistic descriptions based on known concepts. As an evaluation, we use

novel concepts to answer questions and accuracy is the result.

**Training on Licon-T:**

For Licon-T, to generate the relational graph $G$ of concepts, we utilize the topic distribution of concepts, which is labeled manually by experts. The topic of a concept is considered to be its own hypernym concept. And for the loss in Licon-T, we only consider the entailment relationship of concepts , i.e, the relationship between the concept and its topic. The final loss is computed by:

$$L = l - l_e.$$

Moreover, for Licon-T, the known concepts used in the concept learning process are stored in the pre-trained model as commonsense. So we only train the model to learn novel concepts through linguistic descriptions. Besides, to maintain the knowledge stored in the pre-trained model, we initialize the box embeddings of concepts in Licon-T with the value of normal embeddings of concepts as the centers and only randomly initialize the offsets. The training process is the same as for Licon-A and Licon-I.

### B.6 The details in Graph Neural Network

For Graph Neural Network, we use a two-layer graph convolutional network (Kipf and Welling, 2016). For the node $i$ and $j$ in graph $G$, we assign values to $D_{ij}$ according following formula.

$$D_{ij} = \begin{cases} 1 & if\ i = j, \\ 1 & else\ if\ w_i, w_j\ has\ realtionships, \\ 0 & otherwise. \end{cases}$$
$$(1)$$

Then, for the $i$-th node at the $l$-th layer, its hidden state representation denoted as $h_{i,l}^S$, is updated by the following equation:

$$h_{i,l}^S = \sigma(\sum_{j=1}^{n} D_{ij} W_l h_{i,l-1}^S + b_l), \quad (2)$$

where $W_l, b_l$ are learned parameters and $\sigma$ is the ReLU activation function. In practice, the centers and offsets are calculated independently through GCN.

### B.7 The calculation of concept-level entailment

The concept-level entailment, both entailment and contradiction, is calculated thought the overlap values of two concepts $e_{c1} = \{Cen_{c1}, Off_{c1}\}$ and $e_{c2} = \{Cen_{c2}, Off_{c2}\}$.

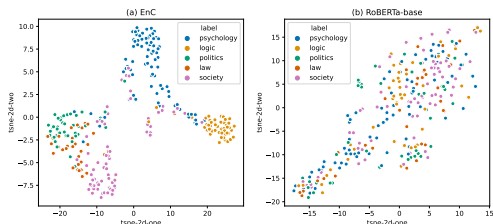

Figure 16: Clustering results of concept embeddings generated by EnC and RoBERTa.

Then we define the lower-bound $Min(e_c)$ and upper-bound $Max(e_c)$ for a box embedding of a concept $c$.

$$Min(e_c) = Cen_c - Off_c \quad (3)$$

$$Max(e_c) = Cen_c + Off_c \quad (4)$$

And the overlap value is calculated through:

$$l[c_1, c_2] = \prod_i \max(M(e_{c1}, e_{c2})_i - m(e_{c1}, e_{c2})_i, 0)$$
$$(5)$$

Where $M(e_{c1}, e_{c2}) = Max(e_{c1}) \vee Max(e_{c2})$ and $m(e_{c1}, e_{c2}) = Min(e_{c1}) \wedge Min(e_{c2})$. $\vee$ and $\wedge$ are element-wise min and max operators. The final loss is then calculated.

## C Performance of EnC on Licon-T

As shown in Table 4, though our proposed method improves the performance of Licon, the results on Licon-T are still unsatisfactory. Because of the following challenges: First, compared to synthetic datasets, relationships in Licon-T are more complex and difficult to categorize into specific classes, such as hypernym or co-hypernym. Therefore, we only label the topic of concepts as hypernym. Second, the linguistic descriptions of novel concepts frequently involve domain-specific terminology that can be hard to understand. Finally, Licon-T demands comprehensive reasoning ability, including deductive, abductive, and analogical reasoning skills.

Due to the aforementioned challenges of Licon-T, we only model the hypernym relationships indicated by the annotated topics, without considering co-hypernym relationships. As a result, the improvement of relationship modeling is not as significant as on synthetic datasets.

Besides, Licon-T requires strong reasoning abilities to determine whether the options belong to the concepts. However, our paper focuses on enhancing the understanding of concepts through box

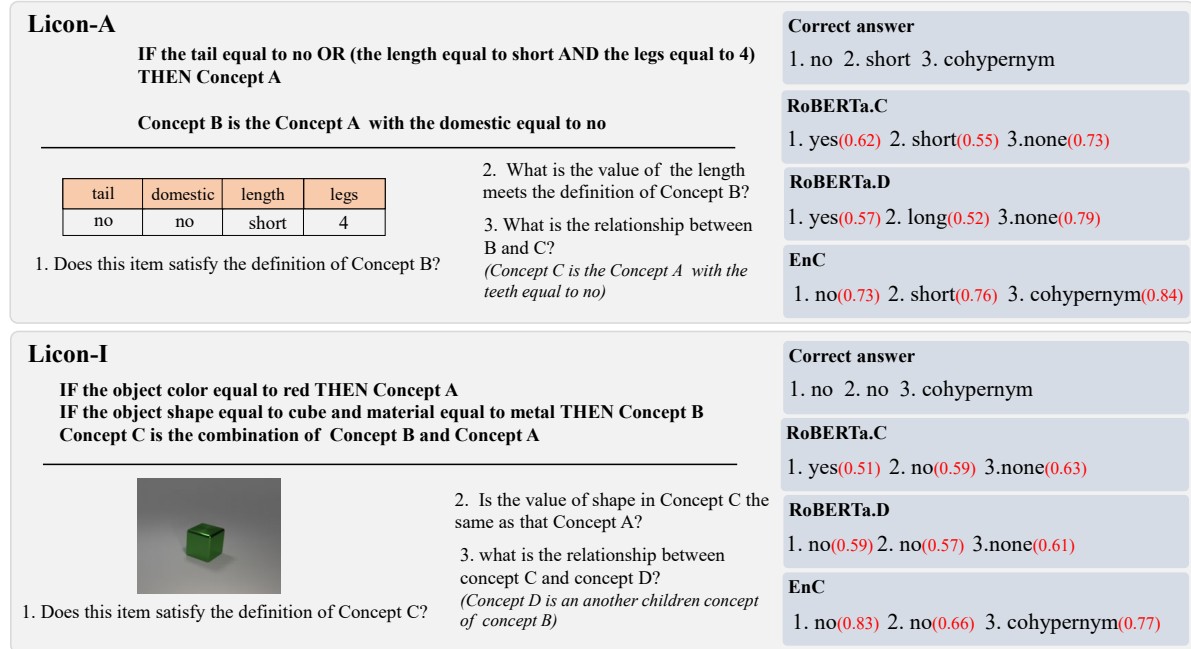

Figure 17: Illustration of typical cases. For a novel concept and its question, we provide the answers and the corresponding probability distribution generated by RoBERTa.C, RoBERTa.D, and EnC.

embeddings, and the emphasis is not on reasoning ability.

Although the performance of EnC on Licon-T is currently unsatisfactory, we conduct cluster experiments using the embeddings generated by EnC and RoBERTa on Licon-T. The results are illustrated in Figure 16. And the results show that the embeddings of concepts with shared topics have shorter distances than those of other concepts. Additionally, the clustering of the embeddings generated by EnC is clearer when compared to those generated by RoBERTa, indicating the effectiveness of EnC. The results show that in the process of learning concepts, EnC is able to link the concepts to their topics through embeddings and therefore leverages the knowledge of known concepts.

## D  Case Study

To further explore the effectiveness of EnC, we present interesting cases in Figure 17. The figure shows the linguistic description of known concepts and novel concepts, the questions about novel concepts, and answers generated by different models.

We can observe that 1) Compared to RoBERTa.C and RoBERTa.D, EnC achieves better performance for both Licon-A and Licon-I. Not only in terms of the accuracy of the answers, but also in terms of the probability distribution of the answers.

2) For RoBERTa.C and RoBERTa.D, the prob-

ability of the answer is slightly higher than 0.5, which indicates that these methods do not actually have a strong ability to learn concepts through the linguistic description. And the answers generated do not have a high degree of certainty. However, EnC is able to generate answers with high certainty.

3) Compared to the baselines, EnC performs particularly well on questions about relationships between concepts. Since the box embeddings maintain the structure of the concepts, EnC is able to answer this type of question. However, for RoBERTa.C and RoBERTa.D, they tend to answer "none" when asked about two concepts that do not occur simultaneously in a sentence.

These findings illustrate the effectiveness of EnC in learning novel concepts based on known concepts through box embeddings.