# OpenReview forum: "Licon: A Diverse, Controllable and Challenging Linguistic Concept Learning Benchmark"
_EMNLP/2023/Conference — EMNLP 2023 Findings_

### Official Review · Reviewer_rgen · 2023-07-30

**Soundness:** 4

**Excitement:**

4: Strong: This paper deepens the understanding of some phenomenon or lowers the barriers to an existing research direction.

**Paper Topic And Main Contributions:**

This paper is about to release a new dataset called Linguistic Concept Learning benchmark (Licon). Different from those datasets where visual information is provided for models to learn concepts, this dataset provides diverse forms of information including plain attributes, images and text, which aims to encourage the learning of abstract concepts which cannot be present by visual information. Furthermore, the Licon dataset has been proved to have the properties of diverse, controllable and challenging with sufficient evaluation and analysis. In addition, this paper also proposes an entailment-based concept learning method (EnC) to model the relationship between concepts, utilizing box embeddings and graph neural networks. This method shows effectiveness on Licon dataset.

**Reasons To Accept:**

This paper releases a new dataset of concept learning. Especially, from my point of view, the task defines in this paper could be used as the instruction to fine-tune or evaluate the LLMs, which may make the LLMs imitate the process of human learning and may improve the generalization of LLMs. Additionally, the dataset itself is diverse, controllable and challenging.

**Reasons To Reject:**

None.

**Reproducibility:**

4: Could mostly reproduce the results, but there may be some variation because of sample variance or minor variations in their interpretation of the protocol or method.

**Reviewer Confidence:**

4: Quite sure. I tried to check the important points carefully. It's unlikely, though conceivable, that I missed something that should affect my ratings.

**Typos Grammar Style And Presentation Improvements:**

In Abstract line 008, I think it should be "Planets". And I think some of your figures could be better illustrated, please make sure you have included all information of the figures in the texts. I guess it is because the limitation of number of pages, some of the import information like the examples of your dataset, should be moved to the main of this paper.

---

> ### Author Rebuttal · Authors · 2023-08-29
>
> Thank you for your suggestions.
> We will redraw the chart to ensure that important information is included in the main part of the paper.

---

### Official Review · Reviewer_y4Hw · 2023-08-04

**Typos Grammar Style And Presentation Improvements:** 1. In Figure-1, Abstract, please corr…
**Soundness:** 3

**Excitement:**

3: Ambivalent: It has merits (e.g., it reports state-of-the-art results, the idea is nice), but there are key weaknesses (e.g., it describes incremental work), and it can significantly benefit from another round of revision. However, I won't object to accepting it if my co-reviewers champion it.

**Paper Topic And Main Contributions:**

In this paper, authors propose a benchmark dataset known as Licon, where concepts in diverse forms or modalities of data are defined by linguistic descriptions. The goal of the benchmark dataset is to perform concept classification, attribute prediction and concept relationship recognition. Authors also propose an approach named EnC which find the relationship among concepts.

**Questions For The Authors:**

1. Does the dataset support annotation of relationship between objects for calculating accuracy and precision?
2. If there are not abstract concepts to learn, what is the issue in visual concept learning?
3. It is not clear from Figure-3, why C is a hierarchy of A,B,E. We can see that A,B,E are subset of C, however they can also exist independently without being in a hierarchy.
4. Licon-I clearly focus on combination of concepts for concept learning as opposed to Licon-A which treats that as hierarchy.
5. How does Licon-T compares itself to other datasets which address reading comprehension task?
6. What is the evaluation metric used in Table-4 results?

**Reasons To Accept:**

1. Language can describe complex relationship between abstract concepts in contrast to visual only representations.
2. Dataset created contain properties such as diverse and controllability.
3. Concept learning method proposed capture relationships quantified by entailment and contradiction.
4. Evaluation is mostly zero-shot by answering multiple choice questions related to the novel concepts which are not part of training sets.

**Reasons To Reject:**

1. Licon dataset is very focused on specific types of concept learning and not present some scalable options for expansion.
2. Specific part of dataset like Licon-A about hierarchical concepts are not particularly seen as hierarchical. They can be treated as subset of each other.
3. The difficulty part of the dataset is not comprehensible as none of the experiments showcase as it claimed. Future experiments were planned with curriculum learning.

**Reproducibility:**

1: Could not reproduce the results here no matter how hard they tried.

**Reviewer Confidence:**

3: Pretty sure, but there's a chance I missed something. Although I have a good feel for this area in general, I did not carefully check the paper's details, e.g., the math, experimental design, or novelty.

---

> ### Author Rebuttal · Authors · 2023-08-29
>
> Thanks for your advice.
> Q1: The Licon dataset is very focused on specific types of concept learning and does not present some scalable options for expansion.
> A1: Our proposal of "linguistic concept learning" is based on the fact that humans are able to learn novel concepts quickly from linguistic descriptions.  In Linguistic Concept Learning, the process of learning concepts relies on textual descriptions. However, known concepts and new concepts themselves can be presented by various types of information, including single-modality and multi-modality.
> For the Licon dataset, we created three sub-datasets in which concepts are presented in different forms, including simple attributes, concrete visual images, and abstract textual definitions. These datasets are controllable and we can test the models with different, scalable hierarchies.
> In the future, we can also extend the concepts to include more types of information, such as audio. Besides, the EnC approach, which focuses on modeling relationships between concepts, can also be easily extended for various types of concepts.
>
> Q2: Clarification about concept hierarchy
> A2: In concept learning, there are relationships of "including" and "being included" among concepts. To make this relationship clearer, following existing methods [1], we use a hierarchical structure to represent the relationship between concepts. In hierarchy, child concepts are generated based on more refined definitions of hypernym concepts and child concepts are the subset of hypernym concepts.
> As shown in Figure 3, for Licon-A, concept B is defined by adding new attribute to A. Therefore, B is the subset of A and A is the hypernym concept of B.
> In Licon-I, concept B is defined by combining the attributes of D and A. Concept B is the subset of both A and D, and concept A and D are both the hypernym concepts of B. Similarly, C is simultaneously the subset of B, E, and further the subset of A, D.
> To sum up, the construction of Licon-A focuses on adding new attributes to existing concepts, while Licon-I concentrates on combining attributes of existing concepts. And the concepts in both datasets maintain a hierarchical structure.
>
> Q3: The difficulty part of the dataset is not comprehensible as none of the experiments showcase as it claimed. Future experiments were planned with curriculum learning.
> A3: The challenges of Licon are mainly from the Licon-T dataset.
> As shown in Figure 5 (a), the reading complexity in Licon-T is quite high and the descriptions are hard to understand. As shown in Figure 5 (b), the distribution of concept topics are wide, increasing the challenge of learning these concepts. Furthermore, we can also observe from Table 4 that the performance of the baselines is significantly below satisfactory, suggesting that Licon-T is challenging.
> In future, we plan to introduce curriculum learning to enhance the understanding ability for complex concepts. Similar to human learning, we'll train the model to learn from easy concepts to complex ones. The planned curriculum includes three stages: (1) simple linguistic concepts, (2) complex options for simple concepts, (3) more complex concepts. With this training method, we expect better performance on Licon-T.
>
> Q4: Does the dataset support annotation of relationship between objects for calculating accuracy and precision?&& What is the evaluation metric used in Table-4 results?
> A4: In Licon's synthetic datasets (Licon-A and Licon-I), we create novel concepts according to the pre-defined relationships between known concepts. Therefore, there is no need to annotate relationships between objects. And for Licon-T, we have annotated topics as hypernyms of concepts for training.
> During evaluation, we utilize questions about the given novel concept (as shown in Figures 10, 11, and 12) to measure the model's understanding of that novel concept. Therefore, we report the accuracy in Tables 2,3 and 4 as the results of the evaluation.
>
> Q5: If there are not abstract concepts to learn, what is the issue in visual concept learning?
> A5: In visual concept learning tasks, concepts are mainly concrete. As shown in Fig. 2 (a), concepts in the visual domain may be related to color, shape, and location, which can be directly represented by images. And the process of learning novel concepts relies on images. As mentioned in [2], "Visual reasoning aims to reason about object properties and their relationships in given images", "such as left, yellow, and cube".
> [2] "FALCON: fast visual concept learning by integrating images, linguistic descriptions, and conceptual relations", ICLR, 2022
>
> Q6: How does Licon-T compares itself to other datasets which address reading comprehension task?
> A6: Reading comprehension mainly focuses on answering questions based on the content of the text, while Linguistic Concept Learning focuses on understanding the concepts and answering the question based on all learned concepts.
> Besides, in reading comprehension, answers are typically derived directly from the provided text. In contrast, Linguistic Concept Learning concentrates on modeling the associations between different concepts, and the answers are not directly sourced from the context.

---

### Official Review · Reviewer_1ug8 · 2023-08-07

**Typos Grammar Style And Presentation Improvements:** None
**Soundness:** 3

**Excitement:**

3: Ambivalent: It has merits (e.g., it reports state-of-the-art results, the idea is nice), but there are key weaknesses (e.g., it describes incremental work), and it can significantly benefit from another round of revision. However, I won't object to accepting it if my co-reviewers champion it.

**Missing References:**

None

**Paper Topic And Main Contributions:**

This paper proposed a dataset called Licon to study the concept learning capability. Concretely, the Licon contains three spilts: Licon-A which contains basic concepts that are for concept learning, and novel concepts for evaluation. Licon-I is similar to Licon-A but with image. And Licon-T is collected from real-world questions for concept learning and reasoning. With all the three datasets, the author proposed an approach to leverage the box embedding and graph neural network to learn concept. The proposed approach achieves higher performance than the baseline approach.

**Questions For The Authors:**

As the Licon-T dataset is used for evaluating human performance. I wonder what is the GPT-3.5 / GPT-4 performance on these datasets? With large language models, such as GPT, Llama, Vicuna, etc., I wonder whether the proposed approach EnC would still be useful and achieves superior performance than those large language models? If yes, what is the reason? If no, then would we still need EnC?

To clarify the excitement section, it is based on the Licon-T and the potential answer of the above question.

**Reasons To Accept:**

1. The Licon dataset is interesting and important, if this is the first dataset proposed for concept learning.
2. The motivation that learning concept can from both visual world and linguistic world is well justified.
3. The Licon-T dataset is quite interesting. Unfortunately, the description of the dataset is quite limited.

**Reasons To Reject:**

The reasons to reject are all for the Licon dataset.

1. After reading the paper, I don't know how to use the Licon-T dataset. How to split the training and the evaluation? As the Licon-T dataset seems not having the separation of basic concept and the novel concept.
2. For Licon-I dataset, I feel it might be little bit redundant, as the main theme of this paper is about learning concepts from linguistic clues. I wonder if this Licon-I fulfills / fits this theme?
3. The Licon-T dataset description might be too limited to conclude whether this dataset is useful or not.

**Reproducibility:**

3: Could reproduce the results with some difficulty. The settings of parameters are underspecified or subjectively determined; the training/evaluation data are not widely available.

**Reviewer Confidence:**

3: Pretty sure, but there's a chance I missed something. Although I have a good feel for this area in general, I did not carefully check the paper's details, e.g., the math, experimental design, or novelty.

---

> ### Author Rebuttal · Authors · 2023-08-28
>
> Thanks for your advice.
>
> Q1: How to split the training and the evaluation for Licon-T?
> A1: For Licon-T, the concept descriptions and questions are randomly divided into training, validation and test sets by 60%:20%:20%.
> We consider the concepts in the training set as known concepts and the concepts in the validation/testing set as novel concepts. In order to establish the relationship between known concepts and novel concepts, we annotate the topic of concepts as the hypernym of concepts, which are shared between known concepts and novel concepts. Then concepts sharing the same topic are considered as  co-hypernyms concepts.
> Therefore, the training and testing process of Licon-T is the same as other datasets.
> For more details, please refer to Appendix B.5 and Appendix C for the detailed training process and the performance of EnC on Licon-T.
>
> Q2: For Licon-I dataset, I feel it might be little bit redundant, as the main theme of this paper is about learning concepts from linguistic clues. I wonder if this Licon-I fulfill / fits this theme?
> A2: We propose to link novel concepts to known concepts through linguistic description, called "linguistic concept learning". As shown in Fig.2, the core of linguistic concept learning is using linguistic descriptions in  to describe relationships between novel concepts and known concepts (i.e., Step 2). As shown in Fig.2(b), for Licon-I, although known concepts are learned through multimodal information (images and their class names), the process of learning novel concepts relies exclusively on linguistic descriptions and maintains themes. Therefore, the Licon-I dataset is an extension of linguistic concept learning in the specific scenario where the concepts contain multimodal information.
>
> Q3: The Licon-T dataset description might be too limited to conclude whether this dataset is useful or not.
> A3: Licon-T is a real-world dataset derived from civil service exams. The descriptions are used to assess the comprehension skills of the average person, and as such, these descriptions
> (1) describe the widely distributed concepts as shown in Figure 5(b).
> (2) have the ability to check the learning ability of the model in the same situation as humans.
> Thus, Licon-T has important applications in testing the learning ability of models in real-world scenarios, and we believe it is useful for training artificial general intelligence.
> And due to space limitations, more discussion of Licon-T has been placed in the Appendix.
>
> Q4: I wonder what is the GPT-3.5 / GPT-4 performance on these datasets?I wonder whether the proposed approach EnC would still be useful and achieves superior performance than those large language models? If yes, what is the reason? If no, then would we still need EnC.
> A4: We tested the performance of Licon-T on GPT-3.5. However, since GPT-3.5 is frequently updated, we did not report this result in the manuscript.
> The GPT-3.5 version is not available as an open source project. This means that we could not directly integrate EnC into GPT-3.5. Therefore, we evaluated the impact of incorporating EnC methods into GPT-3.5 by testing with samples before and after applying the rationale behind EnC (i.e., relating concepts to their basic concepts).
> We randomly selected 50 concepts and their questions to ask GPT-3.5, and the accuracy was 46% (23/50), which is better than the baseline performance in Table-4 but still below 50%.
> Then we tested EnC differently. When we ask a question, we clearly tell GPT-3.5 which topic the concept belongs to, obtaining accuracy rate of 56% (28/50), which suggests that the EnC method, or specifically, the method of relating novel concepts to their basic concepts, is useful.

---

### Meta-Review · Area_Chair_d3X5 · 2023-09-17

**Recommendation:** 4

**Metareview:**

**Summary:**
This paper introduces a new dataset called the Linguistic Concept Learning benchmark (abbreviated as Licon) designed to investigate concept learning capabilities. The Licon dataset comprises three components: Licon-A, which includes basic concepts for concept learning and novel concepts for evaluation; Licon-I, similar to Licon-A but with images; and Licon-T, collected from real-world questions to assess concept learning and reasoning. The diverse information types are intended to facilitate the learning of abstract concepts that cannot be solely represented through visual information. The primary objectives of this benchmark dataset are concept classification, attribute prediction, and concept relationship recognition. Additionally, the paper proposes an entailment-based concept learning method (EnC) that leverages box embeddings and graph neural networks to model relationships between concepts, demonstrating its effectiveness on the Licon dataset.

**Strengths:**
The reviewers unanimously acknowledge the strengths of this work. This paper introduces the Licon dataset, a novel concept learning dataset that stands out for its diversity, controllability, and challenge. The motivation behind incorporating both the visual and linguistic worlds into concept learning is well-founded. Language, unlike visual-only representations, can describe complex relationships between abstract concepts. The proposed concept learning method captures relationships through entailment and contradiction quantification. Notably, the evaluation largely consists of zero-shot tasks, involving multiple-choice questions related to novel concepts not included in the training set. Finally, the tasks defined in this paper have the potential to serve as instructions for fine-tuning or evaluating Large Language Models (LLMs), enabling LLMs to mimic human learning processes and potentially enhancing their generalization capabilities.

**Weaknesses:**
The Licon dataset is overly focused on specific types of concept learning and lacks scalable expansion options. Additionally, specific sections of the dataset, such as Licon-A, which pertains to hierarchical concepts, do not distinctly exhibit hierarchical characteristics, and the description of the Licon-T dataset is somewhat limited. One of the reviewers expressed confusion regarding the usage of the Licon-T dataset, as it does not appear to differentiate between basic and novel concepts. Regarding the Licon-I dataset, it may be considered somewhat redundant, as the primary theme of this paper revolves around learning concepts from linguistic cues. Lastly, the claimed difficulty level of the dataset is not readily discernible, as none of the experiments effectively demonstrate it.

**Author-Reviewer discussion and acknowledgment:**
Reviewers raised various questions and concerns, to which the authors provided responses during the rebuttal and discussion phase. They outlined improvements and engaged in further discussions, particularly with one of the reviewers. Two of the reviewers have responded and acknowledged the authors' arguments.

**Conclusion:**
The paper is well-written; reviewers suggest that the authors rectify just a few typos.

---

### Decision · Program_Chairs · 2023-10-07

**Decision:**

Accept-Findings

**Comment:**

**Summary:**
This paper introduces a new dataset called the Linguistic Concept Learning benchmark (abbreviated as Licon) designed to investigate concept learning capabilities. The Licon dataset comprises three components: Licon-A, which includes basic concepts for concept learning and novel concepts for evaluation; Licon-I, similar to Licon-A but with images; and Licon-T, collected from real-world questions to assess concept learning and reasoning. The diverse information types are intended to facilitate the learning of abstract concepts that cannot be solely represented through visual information. The primary objectives of this benchmark dataset are concept classification, attribute prediction, and concept relationship recognition. Additionally, the paper proposes an entailment-based concept learning method (EnC) that leverages box embeddings and graph neural networks to model relationships between concepts, demonstrating its effectiveness on the Licon dataset.

**Strengths:**
The reviewers unanimously acknowledge the strengths of this work. This paper introduces the Licon dataset, a novel concept learning dataset that stands out for its diversity, controllability, and challenge. The motivation behind incorporating both the visual and linguistic worlds into concept learning is well-founded. Language, unlike visual-only representations, can describe complex relationships between abstract concepts. The proposed concept learning method captures relationships through entailment and contradiction quantification. Notably, the evaluation largely consists of zero-shot tasks, involving multiple-choice questions related to novel concepts not included in the training set. Finally, the tasks defined in this paper have the potential to serve as instructions for fine-tuning or evaluating Large Language Models (LLMs), enabling LLMs to mimic human learning processes and potentially enhancing their generalization capabilities.

**Weaknesses:**
The Licon dataset is overly focused on specific types of concept learning and lacks scalable expansion options. Additionally, specific sections of the dataset, such as Licon-A, which pertains to hierarchical concepts, do not distinctly exhibit hierarchical characteristics, and the description of the Licon-T dataset is somewhat limited. One of the reviewers expressed confusion regarding the usage of the Licon-T dataset, as it does not appear to differentiate between basic and novel concepts. Regarding the Licon-I dataset, it may be considered somewhat redundant, as the primary theme of this paper revolves around learning concepts from linguistic cues. Lastly, the claimed difficulty level of the dataset is not readily discernible, as none of the experiments effectively demonstrate it.

**Author-Reviewer discussion and acknowledgment:**
Reviewers raised various questions and concerns, to which the authors provided responses during the rebuttal and discussion phase. They outlined improvements and engaged in further discussions, particularly with one of the reviewers. Two of the reviewers have responded and acknowledged the authors' arguments.

**Conclusion:**
The paper is well-written; reviewers suggest that the authors rectify just a few typos.